# THINKING WHILE MOVING: DEEP REINFORCEMENT LEARNING WITH CONCURRENT CONTROL

**Ted Xiao**[1], **Eric Jang**[1], **Dmitry Kalashnikov**[1], **Sergey Levine**[1,2], **Julian Ibarz**[1],
**Karol Hausman**[1]*, **Alexander Herzog**[3]*
[1]Google Brain, [2]UC Berkeley, [3]X
{tedxiao, ejang, dkalashnikov, slevine, julianibarz, karolhausman}@google.com,
alexherzog@x.team

## ABSTRACT

We study reinforcement learning in settings where sampling an action from the policy must be done concurrently with the time evolution of the controlled system, such as when a robot must decide on the next action while still performing the previous action. Much like a person or an animal, the robot must think and move at the same time, deciding on its next action before the previous one has completed. In order to develop an algorithmic framework for such concurrent control problems, we start with a continuous-time formulation of the Bellman equations, and then discretize them in a way that is aware of system delays. We instantiate this new class of approximate dynamic programming methods via a simple architectural extension to existing value-based deep reinforcement learning algorithms. We evaluate our methods on simulated benchmark tasks and a large-scale robotic grasping task where the robot must "think while moving". Videos are available at https://sites.google.com/view/thinkingwhilemoving.

## 1 INTRODUCTION

In recent years, Deep Reinforcement Learning (DRL) methods have achieved tremendous success on a variety of diverse environments, including video games (Mnih et al., 2015), zero-sum games (Silver et al., 2016), robotic grasping (Kalashnikov et al., 2018), and in-hand manipulation tasks (OpenAI et al., 2018). While impressive, all of these examples use a *blocking* observe-think-act paradigm: the agent assumes that the environment will remain static while it thinks, so that its actions will be executed on the same states from which they were computed. This assumption breaks in the *concurrent* real world, where the environment state evolves substantially as the agent processes observations and plans its next actions. As an example, consider a dynamic task such as catching a ball: it is not possible to pause the ball mid-air while waiting for the agent to decide on the next control to command. In addition to solving dynamic tasks where blocking models would fail, thinking and acting concurrently can provide benefits such as smoother, human-like motions and the ability to seamlessly plan for next actions while executing the current one.

Despite these potential benefits, most DRL approaches are mainly evaluated in blocking simulation environments. Blocking environments make the assumption that the environment state will not change between when the environment state is observed and when the action is executed. This assumption holds true in most simulated environments, which encompass popular domains such as Atari (Mnih et al., 2013) and Gym control benchmarks (Brockman et al., 2016). The system is treated in a sequential manner: the agent observes a state, freezes time while computing an action, and finally applies the action and unfreezes time. However, in dynamic real-time environments such as real-world robotics, the synchronous environment assumption is no longer valid. After observing the state of the environment and computing an action, the agent often finds that when it executes an action, the environment state has evolved from what it had initially observed; we consider this environment a *concurrent environment*.

In this paper, we introduce an algorithmic framework that can handle concurrent environments in the context of DRL. In particular, we derive a modified Bellman operator for concurrent MDPs and

---
*Indicates equal contribution.

present the minimal set of information that we must augment state observations with in order to recover blocking performance with Q-learning. We introduce experiments on different simulated environments that incorporate concurrent actions, ranging from common simple control domains to vision-based robotic grasping tasks. Finally, we show an agent that acts concurrently in a real-world robotic grasping task is able to achieve comparable task success to a blocking baseline while acting 49% faster.

## 2 RELATED WORK

**Minimizing Concurrent Effects**    Although real-world robotics systems are inherently concurrent, it is sometimes possible to engineer them into approximately blocking systems. For example, using low-latency hardware (Abbeel et al., 2006) and low-footprint controllers (Cruz et al., 2017) minimizes the time spent during state capture and policy inference. Another option is to design actions to be executed to completion via closed-loop feedback controllers and the system velocity is decelerated to zero before a state is recorded (Kalashnikov et al., 2018). In contrast to these works, we tackle the concurrent action execution directly in the learning algorithm. Our approach can be applied to tasks where it is not possible to wait for the system to come to rest between deciding new actions.

**Algorithmic Approaches**    Other works utilize algorithmic modifications to directly overcome the challenges of concurrent control. Previous work in this area can be grouped into five approaches: (1) learning policies that are robust to variable latencies (Tan et al., 2018), (2) including past history such as frame-stacking (Haarnoja et al., 2018), (3) learning dynamics models to predict the future state at which the action will be executed (Firoiu et al., 2018; Amiranashvili et al., 2018), (4) using a time-delayed MDP framework (Walsh et al., 2007; Firoiu et al., 2018; Schuitema et al., 2010), and (5) temporally-aware architectures such as Spiking Neural Networks (Vasilaki et al., 2009; Frémaux et al., 2013), point processes (Upadhyay et al., 2018; Li et al., 2018), and adaptive skip intervals (Neitz et al., 2018). In contrast to these works, our approach is able to (1) optimize for a specific latency regime as opposed to being robust to all of them, (2) consider the properties of the source of latency as opposed to forcing the network to learn them from high-dimensional inputs, (3) avoid learning explicit forward dynamics models in high-dimensional spaces, which can be costly and challenging, (4) consider environments where actions are interrupted as opposed to discrete-time time-delayed environments where multiple actions are queued and each action is executed until completion. The approaches in (5) show promise in enabling asynchronous agents, but are still active areas of research that have not yet been extended to high-dimensional, image-based robotic tasks.

**Continuous-time Reinforcement Learning**    While previously mentioned related works largely operate in discrete-time environments, framing concurrent environments as continuous-time systems is a natural framework to apply. In the realm of continuous-time optimal control, path integral solutions (Kappen, 2005; Theodorou et al., 2010) are linked to different noise levels in system dynamics, which could potentially include latency that results in concurrent properties. Finite differences can approximate the Bellman update in continuous-time stochastic control problems (Munos & Bourgine, 1998) and continuous-time temporal difference learning methods (Doya, 2000) can utilize neural networks as function approximators (Coulom, 2002). The effect of time-discretization (converting continuous-time environments to discrete-time environments) is studied in Tallec et al. (2019), where the advantage update is scaled by the time discretization parameter. While these approaches are promising, it is untested how these methods may apply to image-based DRL problems. Nonetheless, we build on top of many of the theoretical formulations in these works, which motivate our applications of deep reinforcement learning methods to more complex, vision-based robotics tasks.

## 3 VALUE-BASED REINFORCEMENT LEARNING IN CONCURRENT ENVIRONMENTS

In this section, we first introduce the concept of concurrent environments, and then describe the preliminaries necessary for discrete- and continuous-time RL formulations. We then describe the

MDP modifications sufficient to represent concurrent actions and finally, present value-based RL algorithms that can cope with concurrent environments.

The main idea behind our method is simple and can be implemented using small modifications to standard value-based algorithms. It centers around adding additional information to the learning algorithm (in our case, adding extra information about the previous action to a $Q$-function) that allows it to cope with concurrent actions. Hereby, we provide theoretical justification on why these modifications are necessary and we specify the details of the algorithm in Alg. 1.

While concurrent environments affect DRL methods beyond model-free value-based RL, we focus our scope on model-free value-based methods due to their attractive sample-efficiency and off-policy properties for real-world vision-based robotic tasks.

## 3.1 CONCURRENT ACTION ENVIRONMENTS

In *blocking* environments (Figure 4a in the Appendix), actions are executed in a sequential blocking fashion that assumes the environment state does not change between when state is observed and when actions are executed. This can be understood as state capture and policy inference being viewed as instantaneous from the perspective of the agent. In contrast, *concurrent* environments (Figure 4b in the Appendix) do not assume a fixed environment during state capture and policy inference, but instead allow the environment to evolve during these time segments.

## 3.2 DISCRETE-TIME REINFORCEMENT LEARNING PRELIMINARIES

We use standard reinforcement learning formulations in both discrete-time and continuous-time settings (Sutton & Barto, 1998). In the discrete-time case, at each time step $i$, the agent receives state $s_i$ from a set of possible states $\mathcal{S}$ and selects an action $a_i$ from some set of possible actions $\mathcal{A}$ according to its policy $\pi$, where $\pi$ is a mapping from $\mathcal{S}$ to $\mathcal{A}$. The environment returns the next state $s_{i+1}$ sampled from a transition distribution $p(s_{i+1}|s_i, a_i)$ and a reward $r(s_i, a_i)$. The return for a given trajectory of states and actions is the total discounted return from time step $i$ with discount factor $\gamma \in (0, 1]$: $R_i = \sum_{k=0}^{\infty} \gamma^k r(s_{i+k}, a_{i+k})$. The goal of the agent is to maximize the expected return from each state $s_i$. The $Q$-function for a given stationary policy $\pi$ gives the expected return when selecting action $\mathbf{a}$ at state $\mathbf{s}$: $Q^\pi(s, a) = \mathbb{E}[R_i|s_i = s, a_i = a]$. Similarly, the value function gives expected return from state $s$: $V^\pi(s) = \mathbb{E}[R_i|s_i = s]$.

The default blocking environment formulation is detailed in Figure 1a.

## 3.3 VALUE FUNCTIONS AND POLICIES IN CONTINUOUS TIME

For the continuous-time case, we start by formalizing a continuous-time MDP with the differential equation:

$$ds(t) = F(s(t), a(t))dt + G(s(t), a(t))d\beta \tag{1}$$

where $\mathcal{S} = \mathbb{R}^d$ is a set of states, $\mathcal{A}$ is a set of actions, $F : \mathcal{S} \times \mathcal{A} \to \mathcal{S}$ and $G : \mathcal{S} \times \mathcal{A} \to \mathcal{S}$ describe the stochastic dynamics of the environment, and $\beta$ is a Wiener process (Ross et al., 1996). In the continuous-time setting, $ds(t)$ is analogous to the discrete-time $p$, defined in Section 3.2. Continuous-time functions $s(t)$ and $a_i(t)$ specify the state and $i$-th action taken by the agent. The agent interacts with the environment through a state-dependent, deterministic policy function $\pi$ and the return $R$ of a trajectory $\tau = (s(t), a(t))$ is given by (Doya, 2000):

$$R(\tau) = \int_{t=0}^{\infty} \gamma^t r(s(t), a(t))dt, \tag{2}$$

which leads to a continuous-time value function (Tallec et al., 2019):

$$V^\pi(s(t)) = \mathbb{E}_{\tau \sim \pi}[R(\tau)|s(t)]$$
$$= \mathbb{E}_{\tau \sim \pi}\left[\int_{t=0}^{\infty} \gamma^t r(s(t), a(t))dt\right], \tag{3}$$

and similarly, a continuous $Q$-function:

$$Q^\pi(s(t), a, t, H) = \mathbb{E}_p\left[\int_{t'=t}^{t'=t+H} \gamma^{t'-t} r(s(t'), a(t'))dt' + \gamma^H V^\pi(s(t+H))\right], \tag{4}$$

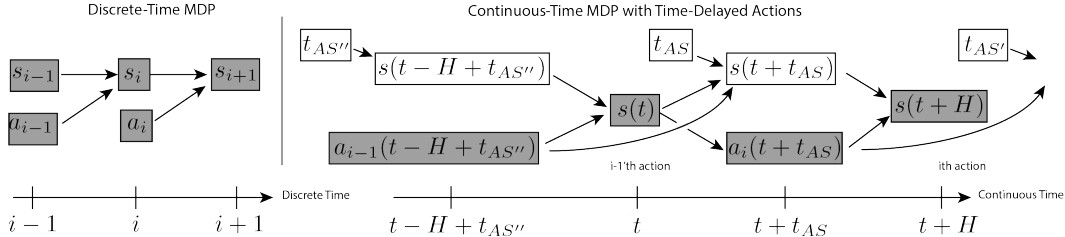

Figure 1: Shaded nodes represent observed variables and unshaded nodes represent unobserved random variables. **(a)**: In "blocking" MDPs, the environment state does not change while the agent records the current state and selects an action. **(b)**: In "concurrent" MDPs, state and action dynamics are continuous-time stochastic processes $s(t)$ and $a_i(t)$. At time $t$, the agent observes the state of the world $s(t)$, but by the time it selects an action $a_i(t + t_{AS})$, the previous continuous-time action function $a_{i-1}(t - H + t_{AS''})$ has "rolled over" to an unobserved state $s(t + t_{AS})$. An agent that concurrently selects actions from old states while in motion may need to interrupt a previous action before it has finished executing its current trajectory.

where $H$ is the constant sampling period between state captures (i.e. the duration of an action trajectory) and $a$ refers to the continuous action function that is applied between $t$ and $t + H$. The expectations are computed with respect to stochastic process $p$ defined in Eq. 1.

## 3.4 Concurrent Action Markov Decision Processes

We consider Markov Decision Processes (MDPs) with concurrent actions, where actions are not executed to full completion. More specifically, concurrent action environments capture system state while the previous action is still executed. After state capture, the policy selects an action that is executed in the environment regardless of whether the previous action has completed, as shown in Figure 4 in the Appendix. In the continuous-time MDP case, concurrent actions can be considered as horizontally translating the action along the time dimension (Walsh et al., 2007), and the effect of concurrent actions is illustrated in Figure 1b. Although we derive Bellman Equations for handling delays in both continuous and discrete-time RL, our experiments extend existing DRL implementations that are based on discrete time.

## 3.5 Value-based Concurrent Reinforcement Learning Algorithms in Continuous and Discrete-Time

We start our derivation from this continuous-time reinforcement learning standpoint, as it allows us to easily characterize the concurrent nature of the system. We then demonstrate that the conclusions drawn for the continuous case also apply to the more commonly-used discrete setting that we then use in all of our experiments.

**Continuous Formulation** In order to further analyze the concurrent setting, we introduce the following notation. As shown in Figure 1b, an agent selects $N$ action trajectories during an episode, $a_1, ..., a_N$, where each $a_i(t)$ is a continuous function generating controls as a function of time $t$. Let $t_{AS}$ be the time duration of state capture, policy inference and any additional communication latencies. At time $t$, an agent begins computing the $i$-th trajectory $a_i(t)$ from state $s(t)$, while concurrently executing the previous selected trajectory $a_{i-1}(t)$ over the time interval $(t - H + t_{AS}, t + t_{AS})$. At time $t + t_{AS}$, where $t \leq t + t_{AS} \leq t + H$, the agent switches to executing actions from $a_i(t)$. The continuous-time $Q$-function for the concurrent case from Eq. 4 can be expressed as following:

$$Q^\pi(s(t), a_{i-1}, a_i, t, H) = \mathbb{E}_p\underbrace{\left[\int_{t'=t}^{t'=t+t_{AS}} \gamma^{t'-t}r(s(t'), a_{i-1}(t'))dt'\right]}_{\text{Executing action trajectory } a_{i-1}(t) \text{ until } t + t_{AS}}$$

$$+ \mathbb{E}_p\underbrace{\left[\int_{t'=t+t_{AS}}^{t'=t+H} \gamma^{t'-t}r(s(t'), a_i(t'))dt'\right]}_{\text{Executing action trajectory } a_i(t) \text{ until } t + H} + \underbrace{\mathbb{E}_p\left[\gamma^H V^\pi(s(t+H))\right]}_{\text{Value function at } t + H} \quad (5)$$

The first two terms correspond to expected discounted returns for executing the action trajectory $a_{i-1}(t)$ from time $(t, t + t_{AS})$ and the trajectory $a_i(t)$ from time $(t + t_{AS}, t + t_{AS} + H)$. We can obtain a single-sample Monte Carlo estimator $\hat{Q}$ by sampling random functions values $p$, which simply correspond to policy rollouts:

$$\hat{Q}^\pi(s(t), a_{i-1}, a_i, t, H) = \int_{t'=t}^{t'=t+t_{AS}} \gamma^{t'-t}r(s(t'), a_{i-1}(t'))dt' +$$

$$\gamma^{t_{AS}}\left[\int_{t'=t+t_{AS}}^{t'=t+H} \gamma^{t'-t-t_{AS}}r(s(t'), a_i(t'))dt' + \gamma^{H-t_{AS}}V^\pi(s(t+H))\right]$$
$$(6)$$

Next, for the continuous-time case, let us define a new concurrent Bellman backup operator:

$$\mathcal{T}_c^*\hat{Q}(s(t), a_{i-1}, a_i, t, t_{AS}) = \int_{t'=t}^{t'=t+t_{AS}} \gamma^{t'-t}r(s(t'), a_{i-1}(t'))dt' +$$

$$\gamma^{t_{AS}}\max_{a_{i+1}}\mathbb{E}_p\hat{Q}^\pi(s(t+t_{AS}), a_i, a_{i+1}, t + t_{AS}, H - t_{AS}). \quad (7)$$

In addition to expanding the Bellman operator to take into account concurrent actions, we demonstrate that this modified operator maintain its contraction properties that are crucial for Q-learning convergence.

**Lemma 3.1.** *The concurrent continuous-time Bellman operator is a contraction.*

*Proof.* See Appendix A.2. □

**Discrete Formulation** In order to simplify the notation for the discrete-time case where the distinction between the action function $a_i(t)$ and the value of that function at time step $t$, $a_i(t)$, is not necessary, we refer to the current state, current action, and previous action as $s_t$, $a_t$, $a_{t-1}$ respectively, replacing subindex $i$ with $t$. Following this notation, we define the concurrent $Q$-function for the discrete-time case:

$$Q^\pi(s_t, a_{t-1}, a_t, t, t_{AS}, H) =$$
$$r(s_t, a_{t-1}) + \gamma^{\frac{t_{AS}}{H}}\mathbb{E}_{p(s_{t+t_{AS}}|s_t, a_{t-1})}Q^\pi(s_{t+t_{AS}}, a_t, a_{t+1}, t + t_{AS}, t_{AS'}, H - t_{AS})$$
$$(8)$$

Where $t_{AS'}$ is the "spillover duration" for action $a_t$ beginning execution at time $t + t_{AS}$ (see Figure 1b). The concurrent Bellman operator, specified by a subscript $c$, is as follows:

$$\mathcal{T}_c^*Q(s_t, a_{t-1}, a_t, t, t_{AS}, H) =$$
$$r(s_t, a_{t-1}) + \gamma^{\frac{t_{AS}}{H}}\max_{a_{t+1}}\mathbb{E}_{p(s_{t+t_{AS}}|s_t, a_{t-1})}Q^\pi(s_{t+t_{AS}}, a_t, a_{t+1}, t + t_{AS}, t_{AS'}, H - t_{AS}).$$
$$(9)$$

Similarly to the continuous-time case, we demonstrate that this Bellman operator is a contraction.

**Lemma 3.2.** *The concurrent discrete-time Bellman operator is a contraction.*

*Proof.* See Appendix A.2. □

We refer the reader to Appendix A.1 for more detailed derivations of the Q-functions and Bellman operators. Crucially, Equation 9 implies that we can extend a conventional discrete-time Q-learning framework to handle MDPs with concurrent actions by providing the Q function with values of $t_{AS}$ and $a_{t-1}$, in addition to the standard inputs $s_t, a_t, t$.

## 3.6 Deep $Q$-Learning with Concurrent Knowledge

While we have shown that knowledge of the concurrent system properties ($t_{AS}$ and $a_{t-1}$, as defined previously for the discrete-time case) is theoretically sufficient, it is often hard to accurately predict $t_{AS}$ during inference on a complex robotics system. In order to allow practical implementation of our algorithm on a wide range of RL agents, we consider three additional features encapsulating *concurrent knowledge* used to condition the $Q$-function: (1) Previous action ($a_{t-1}$), (2) Action selection time ($t_{AS}$), and (3) Vector-to-go ($VTG$), which we define as the remaining action to be executed at the instant the state is measured. We limit our analysis to environments where $a_{t-1}, t_{AS}$, and $VTG$ are all obtainable and $H$ is held constant. See Appendix A.3 for details.

## 4 Experiments

In our experimental evaluation we aim to study the following questions: (1) Is concurrent knowledge defined in Section 3.6, both necessary and sufficient for a $Q$-function to recover the performance of a blocking unconditioned $Q$-function, when acting in a concurrent environment? (2) Which representations of concurrent knowledge are most useful for a $Q$-function to act in a concurrent environment? (3) Can concurrent models improve smoothness and execution speed of a real-robot policy in a realistic, vision-based manipulation task?

### 4.1 Toy First-Order Control Problems

First, we illustrate the effects of a concurrent control paradigm on value-based DRL methods through an ablation study on concurrent versions of the standard Cartpole and Pendulum environments. We use 3D MuJoCo based implementations in DeepMind Control Suite (Tassa et al., 2018) for both tasks. For the baseline learning algorithm implementations, we use the TF-Agents (Guadarrama et al., 2018) implementations of a Deep $Q$-Network agent, which utilizes a Feed-forward Neural Network (FNN), and a Deep $Q$-Recurrent Neutral Network agent, which utilizes a Long Short-Term Memory (LSTM) network. To approximate different difficulty levels of latency in concurrent environments, we utilize different parameter combinations for action execution steps and action selection steps ($t_{AS}$). The number of action execution steps is selected from {0ms, 5ms, 25ms, or 50ms} once at environment initialization. $t_{AS}$ is selected from {0ms, 5ms, 10ms, 25ms, or 50ms} either once at environment initialization or repeatedly at every episode reset. In addition to environment parameters, we allow trials to vary across model parameters: number of previous actions to store, number of previous states to store, whether to use VTG, whether to use $t_{AS}$, $Q$-network architecture, and number of discretized actions. Further details are described in Appendix A.4.1.

To estimate the relative importance of different concurrent knowledge representations, we conduct an analysis of the sensitivity of each type of concurrent knowledge representations to combinations of the other hyperparameter values, shown in Figure 2a. While all combinations of concurrent knowledge representations increase learning performance over baselines that do not leverage this information, the clearest difference stems from including VTG. In Figure 2b we conduct a similar analysis but on a Pendulum environment where $t_{AS}$ is fixed every environment; thus, we do not focus on $t_{AS}$ for this analysis but instead compare the importance of VTG with frame-stacking previous actions and observations. While frame-stacking helps nominally, the majority of the performance increase results from utilizing information from VTG.

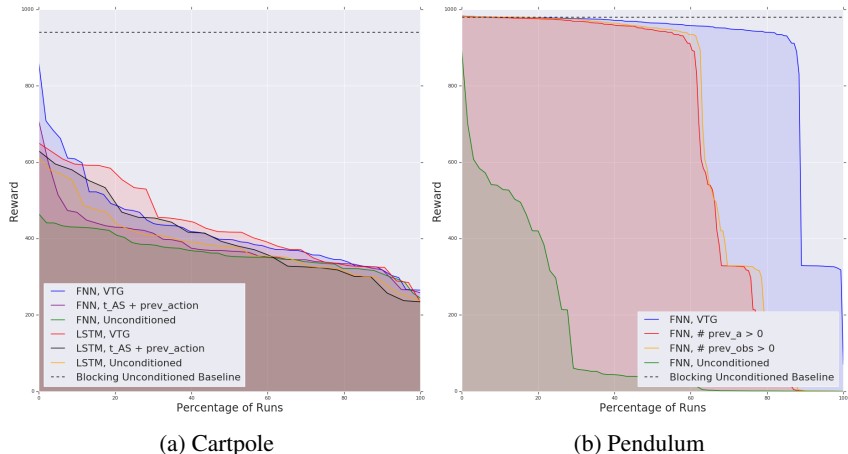

(a) Cartpole        (b) Pendulum

Figure 2: In concurrent versions of Cartpole and Pendulum, we observe that providing the critic with VTG leads to more robust performance across all hyperparameters. **(a)** Environment rewards achieved by DQN with different network architectures [either a feedforward network (FNN) or a Long Short-Term Memory (LSTM) network] and different concurrent knowledge features [Unconditioned, Vector-to-go (VTG), or previous action and $t_{AS}$] on the concurrent Cartpole task for every hyperparameter in a sweep, sorted in decreasing order. **(b)** Environment rewards achieved by DQN with a FNN and different frame-stacking and concurrent knowledge parameters on the concurrent Pendulum task for every hyperparameter in a sweep, sorted in decreasing order. Larger area-under-curve implies more robustness to hyperparameter choices. Enlarged figures provided in Appendix A.5.

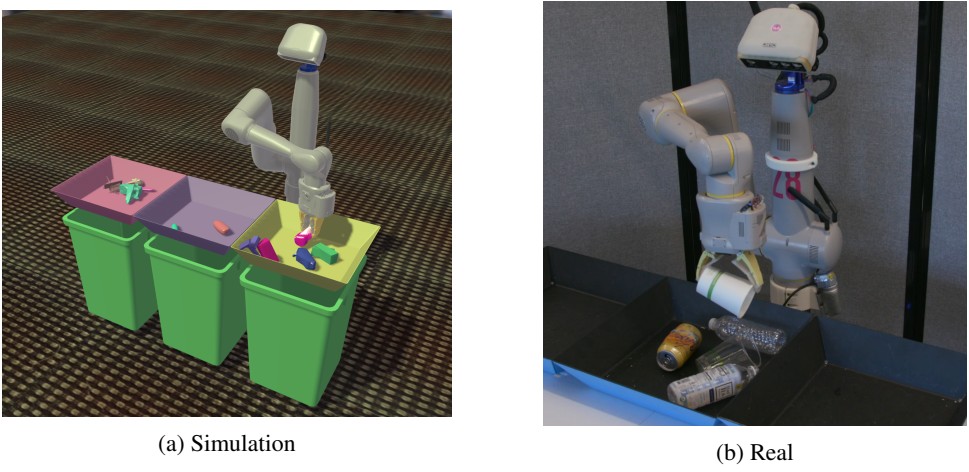

(a) Simulation        (b) Real

Figure 3: An overview of the robotic grasping task. A static manipulator arm attempts to grasp objects placed in bins front of it. In simulation, the objects are procedurally generated.

Table 1: Large-Scale Simulated Robotic Grasping Results

| Blocking Actions | Timestep Penalty | VTG | Previous Action | Grasp Success | Episode Duration | Action Completion |
|---|---|---|---|---|---|---|
| Yes | No | No | No | 92.72% ± 1.10% | 132.09s ±5.70s | **92.33**% ± 1.476% |
| Yes | Yes | No | No | 91.53% ± 1.04% | 120.81s ±9.13s | 89.53% ± 2.267% |
| No | No | No | No | 84.11% ± 7.61% | 122.15s ±14.6s | 43.4% ± 22.41% |
| No | Yes | No | No | 83.77% ± 9.27% | 97.16s ±6.28s | 34.69% ± 16.80% |
| No | Yes | Yes | No | 92.55% ± 4.39% | **82.98s ± 5.74s** | 47.28% ± 14.25% |
| No | Yes | No | Yes | 92.70% ± 1.42% | 87.15s ±4.80s | 50.09% ± 14.25% |
| No | Yes | Yes | Yes | **93.49**% ± **1.04**% | 90.75s ±4.15s | 49.19% ± 14.98% |

## 4.2 CONCURRENT QT-OPT ON LARGE-SCALE ROBOTIC GRASPING

Next, we evaluate scalability of our approach to a practical robotic grasping task. We simulate a 7 DoF arm with an over-the-shoulder camera, where a bin in front of the robot is filled with procedurally generated objects to be picked up by the robot. A binary reward is assigned if an object is lifted off a bin at the end of an episode. We train a policy with QT-Opt (Kalashnikov et al., 2018), a deep $Q$-Learning method that utilizes the cross-entropy method (CEM) to support continuous actions. In the blocking mode, a displacement action is executed until completion: the robot uses a closed-loop controller to fully execute an action, decelerating and coming to rest before observing the next state. In the concurrent mode, an action is triggered and executed without waiting, which means that the next state is observed while the robot remains in motion. Further details of the algorithm and experimental setup are shown in Figure 3 and explained in Appendix A.4.2.

Table 1 summarizes the performance for blocking and concurrent modes comparing unconditioned models against the concurrent knowledge models described in Section 3.6. Our results indicate that the VTG model acting in concurrent mode is able to recover baseline task performance of the blocking execution unconditioned baseline, while the unconditioned baseline acting in concurrent model suffers some performance loss. In addition to the success rate of the grasping policy, we also evaluate the speed and smoothness of the learned policy behavior. Concurrent knowledge models are able to learn faster trajectories: *episode duration*, which measures the total amount of wall-time used for an episode, is reduced by 31.3% when comparing concurrent knowledge models with blocking unconditioned models, even those that utilize a shaped *timestep penalty* that reward faster policies. When switching from blocking execution mode to concurrent execution mode, we see a significantly lower *action completion*, measured as the ratio from executed gripper displacement to commanded displacement, which expectedly indicates a switch to a concurrent environment. The concurrent knowledge models have higher action completions than the unconditioned model in the concurrent environment, which suggests that the concurrent knowledge models are able to utilize more efficient motions, resulting in smoother trajectories. The qualitative benefits of faster, smoother trajectories are drastically apparent when viewing video playback of learned policies[1].

**Real robot results** In addition, we evaluate qualitative policy behaviors of concurrent models compared to blocking models on a real-world robot grasping task, which is shown in Figure 3b. As seen in Table 2, the models achieve comparable grasp success, but the concurrent model is 49% faster than the blocking model in terms of *policy duration*, which measures the total execution time of the policy (this excludes the infrastructure setup and teardown times accounted for in episode duration, which can not be optimized with concurrent actions). In addition, the concurrent VTG model is able to execute smoother and faster trajectories than the blocking unconditioned baseline, which is clear in video playback[1].

## 5 DISCUSSION AND FUTURE WORK

We presented a theoretical framework to analyze concurrent systems where an agent must "think while moving".Viewing this formulation through the lens of continuous-time value-based reinforce-

---

[1]https://sites.google.com/view/thinkingwhilemoving

Table 2: Real-World Robotic Grasping Results.

| Blocking Actions | VTG | Grasp Success | Policy Duration |
|---|---|---|---|
| Yes | No | **81.43**% | 22.60s ±12.99s |
| No | Yes | 68.60% | **11.52s ± 7.272s** |

ment learning, we showed that by considering concurrent knowledge about the time delay $t_{AS}$ and the previous action, the concurrent continuous-time and discrete-time Bellman operators remained contractions and thus maintained $Q$-learning convergence guarantees. While more information than $t_{AS}$ and previous action may be helpful, we showed that $t_{AS}$ and previous action (and different representations of this information) are the sole theoretical requirements for good learning performance. In addition, we introduced Vector-to-go (VTG), which incorporates the remaining previous action to be executed, as an alternative representation for information about the concurrent system that previous action and $t_{AS}$ contain.

Our theoretical findings were supported by experimental results on $Q$-learning models acting in simulated control tasks that were engineered to support concurrent action execution. We conducted large-scale ablation studies on toy task concurrent 3D Cartpole and Pendulum environments, across model parameters as well as concurrent environment parameters. Our results indicated that VTG is the least hyperparameter-sensitive representation, and was able to recover blocking learning performance in concurrent settings. We extended these results to a complex concurrent large-scale sienvironmentmulated robotic grasping task, where we showed that the concurrent models were able to recover blocking execution baseline model success while acting 31.3% faster. We analyzed the qualitative benefits of concurrent models through a real-world robotic grasping task, where we showed that a concurrent model with comparable grasp success as a blocking baseline was able to learn smoother trajectories that were 49% faster.

An interesting topic to explore in future work is the possibility of increased data efficiency when training on off-policy data from various latency regimes. Another natural extension of this work is to evaluate DRL methods beyond value-based algorithms, such as on-policy learning and policy gradient approaches. Finally, concurrent methods may allow robotic control in dynamic environments where it is not possible for the robot to stop the environment before computing the action. In these scenarios, robots must truly think and act at the same time.

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

## A  APPENDIX

### A.1  DEFINING BLOCKING BELLMAN OPERATORS

As introduced in Section 3.5, we define a continuous-time $Q$-function estimator with concurrent actions.

$$\hat{Q}(s(t), a_{i-1}, a_i, t, H) = \int_{t'=t}^{t'=t+t_{AS}} \gamma^{t'-t} r(s(t'), a_{i-1}(t')) dt' + \tag{10}$$

$$\int_{t''=t+t_{AS}}^{t''=t+H} \gamma^{t''-t} r(s(t''), a_i(t'')) dt'' + \gamma^H V(s(t+H)) \tag{11}$$

$$= \int_{t'=t}^{t'=t+t_{AS}} \gamma^{t'-t} r(s(t'), a_{i-1}(t')) dt' + \tag{12}$$

$$\gamma^{t_{AS}} \int_{t''=t+t_{AS}}^{t''=t+H} \gamma^{t''-t-t_{AS}} r(s(t''), a_i(t'')) dt'' + \gamma^H V(s(t+H)) \tag{13}$$

$$= \int_{t'=t}^{t'=t+t_{AS}} \gamma^{t'-t} r(s(t'), a_{i-1}(t')) dt' + \tag{14}$$

$$\gamma^{t_{AS}} \left[ \int_{t''=t+t_{AS}}^{t''=t+H} \gamma^{t''-t-t_{AS}} r(s(t''), a_i(t'')) dt'' + \gamma^{H-t_{AS}} V(s(t+H)) \right] \tag{15}$$

We observe that the second part of this equation (after $\gamma^{t_{AS}}$) is itself a $Q$-function at time $t + t_{AS}$. Since the future state, action, and reward values at $t + t_{AS}$ are not known at time $t$, we take the following expectation:

$$Q(s(t), a_{i-1}, a_i, t, H) = \int_{t'=t}^{t'=t+t_{AS}} \gamma^{t'-t} r(s(t'), a_{i-1}(t')) dt' + \tag{16}$$

$$\gamma^{t_{AS}} \mathbb{E}_s \hat{Q}(s(t), a_i, a_{i+1}, t + t_{AS}, H - t_{AS}) \tag{17}$$

which indicates that the $Q$-function in this setting is not just the expected sum of discounted future rewards, but it corresponds to an expected future $Q$-function.

In order to show the discrete-time version of the problem, we parameterize the discrete-time concurrent $Q$-function as:

$$\hat{Q}(s_t, a_{t-1}, a_t, t, t_{AS}, H) = r(s_t, a_{t-1}) + \gamma^{\frac{t_{AS}}{H}} \mathbb{E}_{p(s_{t+t_{AS}}|s_t, a_{t-1})} r(s_{t+t_{AS}}, a_t) + \tag{18}$$

$$\gamma^{\frac{H}{H}} \mathbb{E}_{p(s_{t+H}|s_{t+t_{AS}}, a_t)} V(s_{t+H}) \tag{19}$$

which with $t_{AS} = 0$, corresponds to a synchronous environment.

Using this parameterization, we can rewrite the discrete-time $Q$-function with concurrent actions as:

$$\hat{Q}(s_t, a_{t-1}, a_t, t, t_{AS}, H) = r(s_t, a_{t-1}) + \gamma^{\frac{t_{AS}}{H}} [\mathbb{E}_{p(s_{t+t_{AS}}|s_t, a_{t-1})} r(s_{t+t_{AS}}, a_t) + \tag{20}$$

$$\gamma^{\frac{H-t_{AS}}{H}} \mathbb{E}_{p(s_{t+H}|st+t_{AS}, a_t)} V(s_{t+H})] \tag{21}$$

$$= r(s_t, a_{t-1}) + \gamma^{\frac{t_{AS}}{H}} \mathbb{E}_{p(s_{t+t_{AS}}|s_t, a_{t-1})} \hat{Q}(s_t, a_t, a_{t+1}, t + t_{AS}, t_{as'}, H - t_{AS}) \tag{22}$$

### A.2  CONTRACTION PROOFS FOR THE BLOCKING BELLMAN OPERATORS

**Proof of the Discrete-time Blocking Bellman Update**

**Lemma A.1.** *The traditional Bellman operator is a contraction, i.e.:*

$$||\mathcal{T}^* \mathcal{Q}_\infty(s, a) - \mathcal{T}^* \mathcal{Q}_\in(s, a)|| \le c ||Q_1(s, a) - Q_2(s, a)||, \tag{23}$$

*where $\mathcal{T}^* \mathcal{Q}(s, a) = r(s, a) + \gamma \max_{a'} \mathbb{E}_p Q(s', a')$ and $0 \le c \le 1$.*

*Proof.* In the original formulation, we can show that this is the case as following:

$$\mathcal{T}^* \mathcal{Q}_1(s,a) - \mathcal{T}^* \mathcal{Q}_2(s,a) \tag{24}$$

$$= r(s,a) + \gamma \max_{a'} \mathbb{E}_p[Q_1(s',a')] - r(s,a) - \gamma \max_{a'} \mathbb{E}_p[Q_2(s',a')] \tag{25}$$

$$= \gamma \max_{a'} \mathbb{E}_p[Q_1(s',a') - Q_2(s',a')] \tag{26}$$

$$\leq \gamma \sup_{s',a'}[Q_1(s',a') - Q_2(s',a')], \tag{27}$$

with $0 \leq \gamma \leq 1$ and $||f||_\infty = \sup_x[f(x)]$. $\square$

Similarly, we can show that the updated Bellman operators introduced in Section 3.5 are contractions as well.

**Proof of Lemma 3.2**

*Proof.*

$$\mathcal{T}_c^* \mathcal{Q}_1(s_t, a_{i-1}, a_i, t, t_{AS}, H) - \mathcal{T}_c^* \mathcal{Q}_2(s_t, a_{i-1}, a_i, t, t_{AS}, H) \tag{28}$$

$$= r(s_t, a_{i-1}) + \gamma^{\frac{t_{AS}}{H}} \max_{a_{i+1}} \mathbb{E}_{p(s_{t+t_{AS}}|s_t, a_{t-1})} Q_1(s_t, a_i, a_{i+1}, t + t_{AS}, t_{AS'}, H - t_{AS}) \tag{29}$$

$$- r(s_t, a_{i-1}) - \gamma^{\frac{t_{AS}}{H}} \max_{a_{i+1}} \mathbb{E}_{p(s_{t+t_{AS}}|s_t, a_{t-1})} Q_2(s_t, a_i, a_{i+1}, t + t_{AS}, t_{AS'}, H - t_{AS}) \tag{30}$$

$$= \gamma^{\frac{t_{AS}}{H}} \max_{a_{i+1}} \mathbb{E}_{p(s_{t+t_{AS}}|s_t, a_{i-1})}[Q_1(s_t, a_i, a_{i+1}, t + t_{AS}, t_{AS'}, H - t_{AS}) - Q_2(s_t, a_i, a_{i+1}, t + t_{AS}, t_{AS'}, H - t_{AS})] \tag{31}$$

$$\leq \gamma^{\frac{t_{AS}}{H}} \sup_{s_t, a_i, a_{i+1}, t+t_{AS}, t_{AS'}, H-t_{AS}}[Q_1(s_t, a_i, a_{i+1}, t + t_{AS}, t_{AS'}, H - t_{AS}) - Q_2(s_t, a_i, a_{i+1}, t + t_{AS}, t_{AS'}, H - t_{AS})] \tag{32}$$

$\square$

**Proof of Lemma 3.1**

*Proof.* To prove that this the continuous-time Bellman operator is a contraction, we can follow the discrete-time proof, from which it follows:

$$\mathcal{T}_c^* \mathcal{Q}_1(s(t), a_{i-1}, a_i, t, t_{AS}) - \mathcal{T}_c^* \mathcal{Q}_2(s(t), a_{i-1}, a_i, t, t_{AS}) \tag{33}$$

$$= \gamma^{t_{AS}} \max_{a_{i+1}} \mathbb{E}_p[Q_1(s(t), a_i, a_{i+1}, t + t_{AS}, H - t_{AS}) - Q_2(s(t), a_i, a_{i+1}, t + t_{AS}, H - t_{AS})] \tag{34}$$

$$\leq \gamma^{t_{AS}} \sup_{s(t), a_i, a_{i+1}, t+t_{AS}, H-t_{AS}}[Q_1(s(t), a_i, a_{i+1}, t + t_{AS}, H - t_{AS}) - Q_2(s(t), a_i, a_{i+1}, t + t_{AS}, H - t_{AS})] \tag{35}$$

$\square$

### A.3 CONCURRENT KNOWLEDGE REPRESENTATION

We analyze 3 different representations of concurrent knowledge in discrete-time concurrent environments, described in Section 3.6. Previous action $a_{t-1}$ is the action that the agent executed at the previous timestep. Action selection time $t_{AS}$ is a measure of how long action selection takes, which can be represented as either a categorical or continuous variable; in our experiments, which take advantage of a bounded latency regime, we normalize action selection time using these known bounds. Vector-to-go *VTG* is a feature that combines $a_{t-1}$ and $s_t$ by encoding the remaining amount of $a_{t-1}$ left to execute. See Figure 5 for a visual comparison.

We note that $a_{t-1}$ is available across the vast majority of environments and it is easy to obtain. Using $t_{AS}$, which encompasses state capture, communication latency, and policy inference, relies

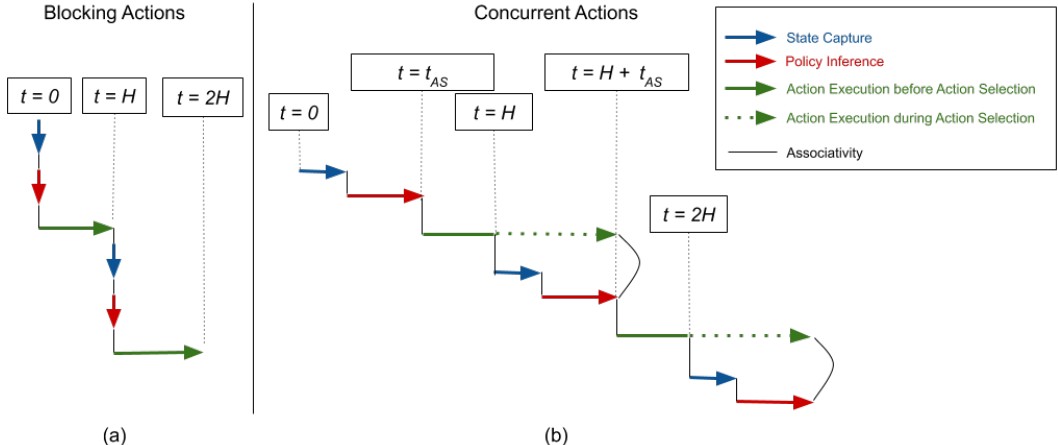

Figure 4: The execution order of different stages are shown relative to the sampling period $H$ as well as the latency $t_{AS}$. (a): In "blocking" environments, state capture and policy inference are assumed to be instantaneous. (b): In "concurrent" environments, state capture and policy inference are assumed to proceed concurrently to action execution.

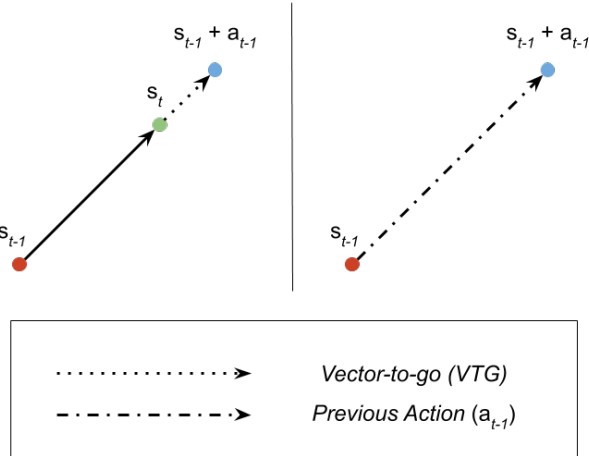

Figure 5: Concurrent knowledge representations can be visualized through an example of a 2-D pointmass discrete-time toy task. Vector-to-go represents the remaining action that may be executed when the current state $s_t$ is observed. Previous action represents the full commanded action from the previous timestep.

on having some knowledge of the concurrent properties of the system. Calculating $VTG$ requires having access to some measure of action completion at the exact moment when state is observed. When utilizing a first-order control action space, such as joint angle or desired pose, $VTG$ is easily computable if proprioceptive state is measured and synchronized with state observation. In these cases, VTG is an alternate representation of the same information encapsulated by $a_{t-1}$ and the current state.

## A.4 Experiment Implementation Details

### A.4.1 Cartpole and Pendulum Ablation Studies

Here, we describe the implementation details of the toy task Cartpole and Pendulum experiments in Section 4.1.

For the environments, we use the 3D MuJoCo implementations of the `Cartpole-Swingup` and `Pendulum-Swingup` tasks in DeepMind Control Suite (Tassa et al., 2018). We use discretized action spaces for first-order control of joint position actuators. For the observation space of both tasks, we use the default state space of ground truth positions and velocities.

For the baseline learning algorithms, we use the TensorFlow Agents (Guadarrama et al., 2018) implementations of a Deep $Q$-Network agent, which utilizes a Feed-forward Neural Network (FNN), and a Deep $Q$-Recurrent Neutral Network agent, which utilizes a Long Short-Term Memory (LSTM) network. Learning parameters such as `learning_rate`, `lstm_size`, and `fc_layer_size` were selected through hyperparameter sweeps.

To approximate different difficulty levels of latency in concurrent environments, we utilize different parameter combinations for action execution steps and action selection steps ($t_{AS}$). The number of action execution steps is selected from {0ms, 5ms, 25ms, or 50ms} once at environment initialization. $t_{AS}$ is selected from {0ms, 5ms, 10ms, 25ms, or 50ms} either once at environment initialization or repeatedly at every episode reset. The selected $t_{AS}$ is implemented in the environment as additional physics steps that update the system during simulated action selection.

Frame-stacking parameters affect the observation space by saving previous observations and actions. The number of previous actions to store as well as the number of previous observations to store are independently selected from the range $[0, 4]$. Concurrent knowledge parameters, as described in Section 4, include whether to use VTG and whether to use $t_{AS}$. Including the previous action is already a feature implemented in the frame-stacking feature of including previous actions. Finally, the number of actions to discretize the continuous space to is selected from the range $[3, 8]$.

### A.4.2 Large Scale Robotic Grasping

**Simulated Environment**   We simulate a 7 DoF arm with an over-the-shoulder camera (see Figure 3a). A bin in front of the robot is filled with procedurally generated objects to be picked up by the robot and a sparse binary reward is assigned if an object is lifted off a bin at the end of an episode. States are represented in form of RGB images and actions are continuous Cartesian displacements of the gripper 3D positions and yaw. In addition, the policy commands discrete gripper open and close actions and may terminate an episode. In blocking mode, a displacement action is executed until completion: the robot uses a closed loop controller to fully execute an action, decelerating and coming to rest before observing the next state. In concurrent mode, an action is triggered and executed without waiting, which means that the next state is observed while the robot remains in motion. It should be noted that in blocking mode, action completion is close to $100\%$ unless the gripper moves are blocked by contact with the environment or objects; this causes average blocking mode action completion to be lower than $100\%$, as seen in Table 1.

**Real Environment**   Similar to the simulated setup, we use a 7 DoF robotic arm with an over-the-shoulder camera (see Figure 3b). The main difference in the physical setup is that objects are selected from a set of common household objects.

**Algorithm**   We train a policy with QT-Opt (Kalashnikov et al., 2018), a Deep $Q$-Learning method that utilizes the Cross-Entropy Method (CEM) to support continuous actions. A Convolutional Neural Network (CNN) is trained to learn the $Q$-function conditioned on an image input along with

a CEM-sampled continuous control action. At policy inference time, the agent sends an image of the environment and batches of CEM-sampled actions to the CNN $Q$-network. The highest-scoring action is then used as the policy's selected action. Compared to the formulation in Kalashnikov et al. (2018), we also add a *concurrent knowledge* feature of VTG and/or previous action $a_{t-1}$ as additional input to the $Q$-network. Algorithm 1 shows the modified QT-Opt procedure.

---

**Algorithm 1:** QT-Opt with Concurrent Knowledge

---

Initialize replay buffer $D$;
Initialize random start state and receive image $o_0$;
Initialize concurrent knowledge features $c_0 = [VTG_0 = 0, a_{t-1} = 0, t_{AS} = 0]$;
Initialize environment state $s_t = [o_0, c_0]$;
Initialize action-value function $Q(s, a)$ with random weights $\theta$;
Initialize target action-value function $\hat{Q}(s, a)$ with weights $\hat{\theta} = \theta$;
**while** *training* **do**
    **for** *t = 1, T* **do**
        Select random action $a_t$ with probability $\epsilon$, else $a_t = \mathbf{CEM}(Q, s_t; \theta)$;
        Execute action in environment, receive $o_{t+1}$, $c_t$, $r_t$;
        Process necessary concurrent knowledge features $c_t$, such as $VTG_t$, $a_{t-1}$, or $t_{AS}$;
        Set $s_{t+1} = [o_{t+1}, c_t]$;
        Store transition $(s_t, a_t, s_{t+1}, r_t)$ in $D$;
        **if** *episode terminates* **then**
            Reset $s_{t+1}$ to a random reset initialization state;
            Reset $c_{t+1}$ to 0;
        **end**
        Sample batch of transitions from $D$;
        **for** *each transition $(s_i, a_i, s_{i+1}, r_i)$ in batch* **do**
            **if** *terminal transition* **then**
                $y_i = r_i$;
            **else**
                Select $\hat{a}_{i+1} = \mathbf{CEM}(\hat{Q}, s_i; \hat{\theta})$;
                $y_i = r_i + \gamma \hat{Q}(s_{i+1}, \hat{a}_{i+1})$;
            **end**
            Perform **SGD** on $(y_i - Q(s_i, a_i; \theta))^2$ with respect to $\theta$;
        **end**
        Update target parameters $\hat{Q}$ with $Q$ and $\theta$ periodically;
    **end**
**end**

---

For simplicity, the algorithm is described as if run synchronously on a single machine. In practice, episode generation, Bellman updates and Q-fitting are distributed across many machines and done asynchronously; refer to (Kalashnikov et al., 2018) for more details. Standard DRL hyperparameters such as random exploration probability ($\epsilon$), reward discount ($\gamma$), and learning rate are tuned through a hyperparameter sweep. For the time-penalized baselines in Table 1, we manually tune a *timestep penalty* that returns a fixed negative reward at every timestep. Empirically we find that a timestep penalty of $-0.01$, relative to a binary sparse reward of $1.0$, encourages faster policies. For the non-penalized baselines, we set a timestep penalty of $-0.0$.

## A.5 FIGURES

See Figure 6 and Figure 7.

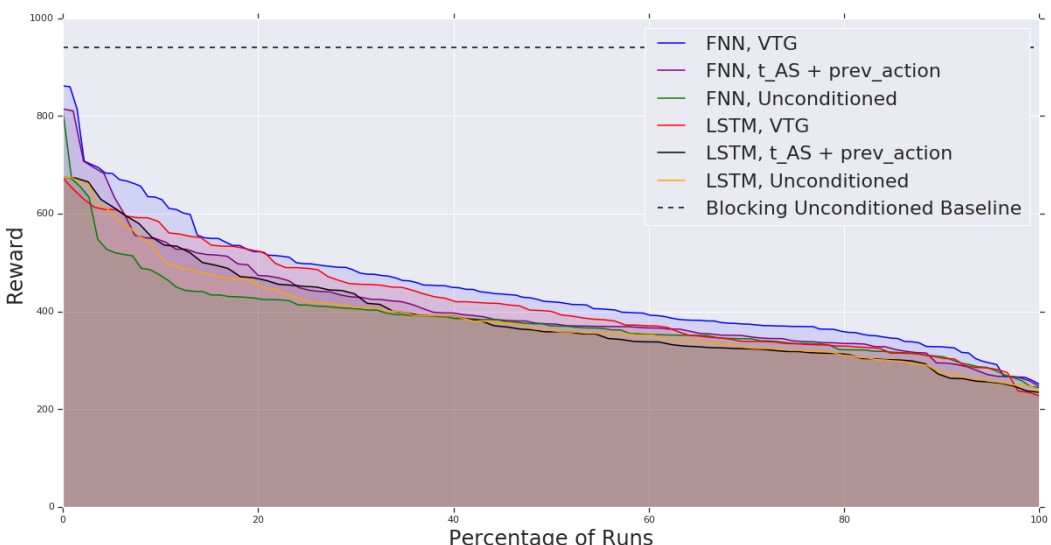

Figure 6: Environment rewards achieved by DQN with different network architectures [either a feedforward network (FNN) or a Long Short-Term Memory (LSTM) network] and different concurrent knowledge features [Unconditioned, vector-to-go (VTG), or previous action and $t_{AS}$] on the concurrent Cartpole task for every hyperparameter in a sweep, sorted in decreasing order. Providing the critic with VTG information leads to more robust performance across all hyperparameters. This figure is a larger version of 2a.

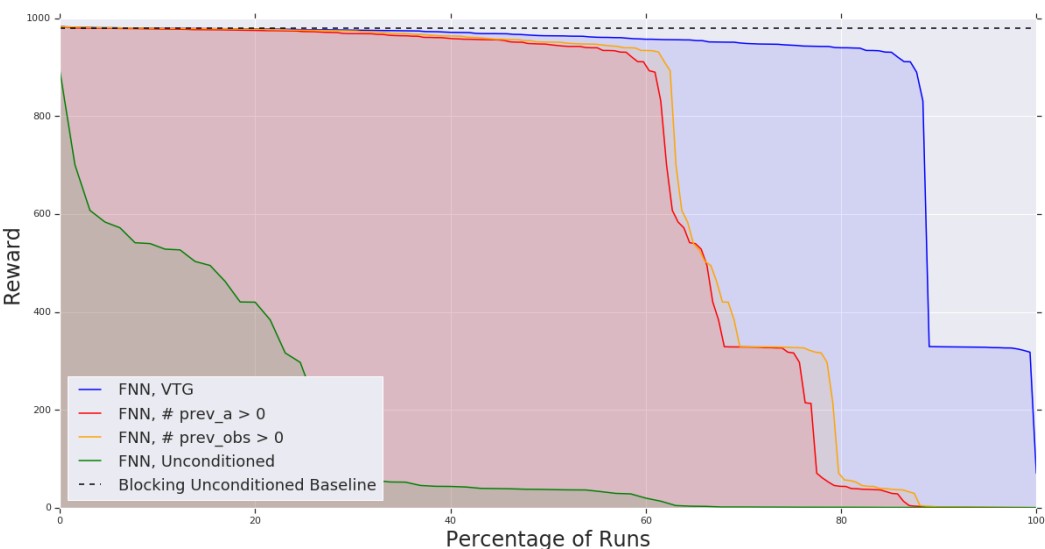

Figure 7: Environment rewards achieved by DQN with a FNN and different frame-stacking and concurrent knowledge parameters on the concurrent Pendulum task for every hyperparameter in a sweep, sorted in decreasing order.

