# OpenReview forum: "Thinking While Moving: Deep Reinforcement Learning with Concurrent Control"
_ICLR.cc/2020/Conference — Accept (Poster)_

### Official Review · AnonReviewer3 · 2019-10-22
**Official Blind Review #3**

**Rating:** 6

**Review:**

The paper tackles the problem of making decisions for the next action while still engaged in doing the previous actions. Such a delay could either be part of the design (like a robot deciding the next action before its actors and motors have come to full rest after the current action) or an artefact of the delays inherent in the system (i.e. latency induced by calculations or latency of sensors). The paper shows how to model such delays within the Q-learning framework, show that their modelling preserves desirable contraction property of the Bellman update operator, and put their model into practice by an extensive set of experiments: learning policies for several simulated and a real-world setting.

The authors claim that that addition of this "delay" does not hinder the performance much of the RL method is given sufficient "context" about the delay, i.e., given extra features as input in order to learn to compensate for it. The writing of the paper is lucid and sufficient background is provided to make the paper self-sufficient in its explanations.

However, there are some reasons which do not allow me to fully support the paper's acceptance.

The changes made to the basic Q-learning setup, albeit novel and with desirable properties, in my opinion, are (i) theoretically relatively straight forward, (ii) are not expressive enough to capture the problem in its full generality (explained later), and (iii) need more empirical justification with problems where their modification is indeed indispensable. The authors touch on several different research areas cursorily (viz. continuous reinforcement learning, Bellman contractions, feature engineering) while providing grounds for their idea, but in the end return to the familiar domain of discrete Q-learning with semi-hand-crafted (though theoretically motivated) features where the latency of actions can take a set of fixed values and the state is sampled at fixed intervals.

If the actions are continuous, then could method from Doya (2000) directly be used to solve these problems? Can the value-based models which he describes be augmented and extensions developed which build on Lemma 3.1 instead of the well-trodden ground of Lemma 3.2? Especially, if one of the objectives which the authors claim their policies are better is "policy duration", then the absence of purely continuous policies is particularly egregious. Further, reducing the policy duration seems like an independent objective which perhaps can be used for reward shaping for the traditional policy methods, which will also lead to different baselines.

The authors explicitly say that their method focuses on "optimizing for a specific latency regime as opposed to being robust to all of them;" and that they explicitly avoid learning forward models by including additional features. However, the advantages of placing such restrictions on the design space are unclear at best. Would it be the case that the high-dimensional methods will fail in this setting? Are there theoretical advantages to working on limiting the attention to known latency regimes? I suspect that the authors have concrete reasons for making these design decisions, but these do not come across in the paper in the writing, or by means of additional baselines.

As an example of a different approach towards the problem, which the authors overlook in their related work section, is that of learning with spiking neurons and point processes. These areas of research have also been interested in problems of the "thinking while moving" nature: that of reinforcement learning in the context of neurons where the neurons act by means of spikes in response to the environment and other "spikes" [1, 2]. More recently, with point processes, methods have been developed to attain truly asynchronous action and state updates [3, 4]. A differently motivated work which ends up dealing with similar problems is in the direction of adaptive skip intervals [5], where the network also chooses the "latency" in the discrete sense. Adding such related work would help better contextualize this paper.

Some other ways the authors can improve the paper are (in no particular order):

 - The description of the Vector-to-go is insufficient; some concrete examples will help.
 - The results of the simulated experiments are given in the form of distributions and it is very difficult to discern the effect of individual features in Figure 1. Additionally, due to missing error bars, or other measures of uncertainty, the claim that the performance of models with and without the delayed-actions is comparable to the blocking setting seems tenuous at best, just looking at the rewards.
 - In particular, for the real experiments, we need more details about the experiment runs to determine why the performance of the policies in the real world is so vastly different. Could the authors describe why the gap can be completely covered through simulations but not in the real world?

[1]: Vasilaki, Eleni, et al. "Spike-based reinforcement learning in continuous state and action space: when policy gradient methods fail." PLoS computational biology 5.12 (2009): e1000586.
[2]: Frémaux, Nicolas, Henning Sprekeler, and Wulfram Gerstner. "Reinforcement learning using a continuous time actor-critic framework with spiking neurons." PLoS computational biology 9.4 (2013): e1003024.
[3]: Upadhyay, Utkarsh, Abir De, and Manuel Gomez Rodriguez. "Deep reinforcement learning of marked temporal point processes." Advances in Neural Information Processing Systems. 2018.
[4]: Li, Shuang, et al. "Learning temporal point processes via reinforcement learning." Advances in Neural Information Processing Systems. 2018.
[5]: Neitz, Alexander, et al. "Adaptive skip intervals: Temporal abstraction for recurrent dynamical models." Advances in Neural Information Processing Systems. 2018.

**Experience Assessment:**

I have published one or two papers in this area.

**Review Assessment: Checking Correctness Of Derivations And Theory:**

I carefully checked the derivations and theory.

**Review Assessment: Checking Correctness Of Experiments:**

I assessed the sensibility of the experiments.

**Review Assessment: Thoroughness In Paper Reading:**

I read the paper thoroughly.

---

> ### Author Response · Authors · 2019-11-13
> **Response to Blind Review #3**
>
> We thank the reviewer for their constructive feedback and exceptionally detailed review.
>
> We agree with the reviewer that although our theoretical justification is based on continuous-time RL and discusses a general framework for handling delays in Q-learning (continuous or discrete), our actual experiments return to the “well-trodden” regime of discrete-time RL with an auxiliary VTG input to the critic network. Regardless of whether the setting is continuous or discrete time RL, the problem of dealing with delays in RL persists. To the best of our knowledge, most of the SOTA DRL implementations are based on discrete-time RL formulations. We are not aware of any image-based DRL results that use a continuous-time RL formulation. While that may be an interesting avenue, we believe this is outside of the scope of our work and believe our method to adapt discrete methods to handle delays is another way to approach the problem. We clarified this in Section 2.
>
> We agree with the reviewer that since “policy duration” is a quantitative metric that we use to support our claim of faster learned trajectories, a reasonable baseline method should incorporate this optimization goal. The baseline model we compare against penalizes slower policies that take more episode steps through reward discount gamma as well as an timestep penalty, a hyperparameter that returns a fixed negative reward every timestep. This timestep penalty was tuned through a hyperparameter search, and is described in further detail in the Appendix. Additionally, in Table 1 we add two baselines that do not utilize this reward penalty.
>
> We also thank the reviewer for suggesting that we clarify the motivations behind restricting the design space. We focus our study on model-free methods because image-based model-based methods, such as video prediction models, are challenging to learn and an active area of research that is tangential to our main focus of studying concurrent environments. We limit our environments to known latency regimes because this is motivated by real-world robotics setups, where latencies can often be constrained within known upper bounds.
>
> We appreciate the reviewer introducing additional related work and improvements that would help contextualize our contribution. These are exciting research directions that we think show much promise for when they are applied to vision-based robotic control tasks. We added a description of spiking neurons, point processes, and adaptive skip intervals in Section 2.
>
> "1. The description of the Vector-to-go is insufficient."
> Thank you for this suggestion. We have clarified the description of concurrent knowledge representations with a section in the appendix as well as with Figure 5.
>
> "2. The results of the simulated experiments are given in the form of distributions and it is very difficult to discern the effect of individual features in Figure 1. Additionally, due to missing error bars, or other measures of uncertainty, the claim that the performance of models with and without the delayed-actions is comparable to the blocking setting seems tenuous at best."
> We assume the reviewer is referring to Figure 2, not Figure 1, as our robotic grasping experiments do indeed report confidence intervals computed over multiple random seeds or real-world evaluations. Figure 2 is a hyperparameter sensitivity plot obtained by performing a hyperparameter tuning experiment across many training runs of the CartPole and Pendulum control tasks. The hyperparameter configurations are then sorted from best to worst, with the X axis plotting the sorted rank of the experiment. One can interpret the entire plot as a distribution over returns over N experiments, where shorter-tailed distributions imply that the method more “robust”. Larger area-under-curve means that obtaining good performance is less sensitive to choice of hyperparameters (which is crucial for getting RL algorithms to work on real robots, where sample complexity is prohibitive).
>
> Because this computationally expensive hyperparameter optimization procedure does not yield multiple i.i.d. experiments w.r.t a single hyperparameter configuration (for computational efficiency), we cannot estimate per-experiment uncertainty from this dataset as commonly done in RL.
>
> "3. Could the authors describe why the gap can be completely covered through simulations but not in the real world?"
> Thank you for this suggestion. To be completely frank, we are not sure exactly why the large-scale grasping success could be covered in the simulated but not in the real world. However, real-world robotic tasks are difficult and sensitive to many parameters. Given that, we still felt it important to report the full results. We also added more experiment details in the Appendix.
>
> Finally, we would like to thank the reviewer for summarizing the main concerns. We felt these insightful comments could be useful to other reviewers, and added our response to the general comment.

---

### Official Review · AnonReviewer1 · 2019-10-26
**Official Blind Review #1**

**Rating:** 6

**Review:**

This paper considers the theoretically interesting and practically important problem of concurrent deep reinforcement learning (DRL), i.e., DRL in which the agent has to decide the next action while performing the previous one. This introduces several significant challenges, including delays/latency and interruption of an on-going action. To address this issue, this paper proposes to consider the continuous time formulation of the concurrent control problem, derive a continuous-time Bellman equation for the concurrent control scenarios, and then derive its discrete-time counterpart. Contraction properties are shown for both the continuous-time and discrete-time concurrent Bellman equations, and a value-based DRL algorithm based on the concurrent Bellman equations is proposed and tested on a few tasks.

The high level idea of this paper is very interesting and attractive, and in particular, the introduction of continuous-time reinforcement learning is novel. In addition, the numerical experiments do show that a consistently improved performance of the proposed approach on both synthetic and more real-world robotic control tasks. However, there are several significant issues about technical clarity or even correctness in this paper, which I elaborate below:

1. The settings in sections 3.1 and 3.2 are not clear. In particular, for 3.1, the author may want to specify the policy clearly, including whether it is stationary or non-stationary. And in addition, Q and V functions should either come with a \pi superscript, indicating which policy they use, or a \star superscript to indicate optimality. Section 3.2 does not make sense to me in general. It is not clear what the index i and the state value s_i(t) are. And it is not clear why we need to differentiate between values of states/actions and the functions themselves. The trajectory \tau is also not clearly defined. The authors need to make these much more clear, and should clearly state the main setting/model that the paper is considering (which seems to be the concurrent discrete-time case, but also not very clear to me).

2. The explanation of concurrent actions in continuous and discrete time is not clear. In particular, Section 3.3 only speaks of the settings on a high level, and only brief explanations are given in Figure 1b and the beginning of Section 3.4. Since the concurrent action setting is the central theme in this paper, I think a much more formal explanation should be given about how the system proceeds, instead of just a graphical example illustration. In addition, the concurrent actions in discrete-time setting part is not even clearly mentioned (but is stated in the title of Section 3.3 and discussed subsequently). The authors may also want to explain clearly what the episode is at the beginning of Section 3.4.

3. The concurrent Bellman equation does not make much sense to me. In particular, I think to define the optimal Q function, the bellman equation (7) and (9) should have a \max operator included. Otherwise, it is only for policy evaluation. Since the authors didn't clearly specify what the exact algorithm they are using (apart from a brief explanation by words in Section 3.5), I'm not sure whether I'm missing anything or not. But the authors should definitely include a algorithm frame at least in the appendix, to clearly specify which of and how the concurrent Bellman equations are applied in their algorithm.

So in sum, although I think the paper is interesting and novel on the high level, I don't think it's ready for publishing.

############ post rebuttal comment ############
After reading the authors' rebuttal and the modified version of the paper, I think most of my concerns have been correctly addressed. So I decide to improve my score to weak accept.

**Experience Assessment:**

I have read many papers in this area.

**Review Assessment: Checking Correctness Of Derivations And Theory:**

I assessed the sensibility of the derivations and theory.

**Review Assessment: Checking Correctness Of Experiments:**

I assessed the sensibility of the experiments.

**Review Assessment: Thoroughness In Paper Reading:**

I read the paper at least twice and used my best judgement in assessing the paper.

---

> ### Author Response · Authors · 2019-11-13
> **Response to Blind Review #1**
>
> Thank you for a thorough review. Please see our responses to your comments below:
>
> "1. The settings in sections 3.1 and 3.2 are not clear."
> We agree with the reviewer that the clarity of the setup and the method can be improved. We addressed the reviewer’s comments in the updated version of the paper. In particular, we: i) added the missing definitions (e.g. trajectory \tau), ii) clarified the exact problem setup considered, iii) added missing superscripts to the notation and iv) simplified the notation by removing the distinction between bolded and unbolded symbols.
>
> "2. The explanation of concurrent actions in continuous and discrete time is not clear."
> Thank you for this excellent suggestion. We added a section (i.e. Section 3.1) and illustrative figure (i.e. Figure 4) that hopefully clarify the main aspect of the paper. We also added the definition of the episode as suggested by the reviewer.
>
> "3. The concurrent Bellman equation does not make much sense to me. "
> Thank you for your suggestions. We updated the manuscript to provide the details about the exact algorithm that was used in the experiments, which hopefully clarifies how the introduced method fits into a bigger robot-learning framework. We also added an algorithm frame in the Appendix that should allow readers to fully understand the contribution of this paper. We also applied the changes to the concurrent Bellman operator that include taking the maximum over actions.

---

> > ### Comment · AnonReviewer1 · 2019-11-13
> > **Rating improved**
> >
> > Thanks for the efforts taken to address my concerns. I have checked the updated draft and I think most of my major concerns have been addressed. And I have improved my rating accordingly. Thanks.

---

### Official Review · AnonReviewer2 · 2019-10-30
**Official Blind Review #2**

**Rating:** 6

**Review:**

This paper proposes a first step in the direction of 'real-world relevant RL approaches' in the sense of considering environments that don't halt their execution until an agent has finished its optimal action computation and execution but actually just go on being an environment. For this, the notion of a concurrent action is introduced.

The paper focuses on value-based RL approaches. It introduces modifications to the classical MDP formulation such that concurrent actions can be handled. From a theoretical perspective the resulting Bellman operators (for both continuous and discrete time) remain contractions and thus maintain q-learning convergence guarantees. Qlearning models are adopted to support concurrent actions and the experiments  demonstrate that the suggested enhancements are working well.

**Experience Assessment:**

I do not know much about this area.

**Review Assessment: Checking Correctness Of Derivations And Theory:**

I did not assess the derivations or theory.

**Review Assessment: Checking Correctness Of Experiments:**

I assessed the sensibility of the experiments.

**Review Assessment: Thoroughness In Paper Reading:**

I made a quick assessment of this paper.

---

> ### Author Response · Authors · 2019-11-13
> **Response to Blind Review #2**
>
> We thank the reviewer for their positive review and an accurate summary of the paper.

---

### Author Response · Authors · 2019-11-13
**General Summary of Changes**

We thank the reviewers for insightful and useful suggestions. We have incorporated feedback into our manuscript and uploaded a new draft.

The writing changes are highlighted in red text.

The main changes we made are:
- More clearly specifying our contribution and the relationship of our work to continuous-time RL methods (Section 3 and Section 3.4)
- Simplifying notation in the derivations (New Section 3, Section A.1, Section A.2)
- Introducing simulated robotic grasping baselines that incorporate a timestep penalty that encourages faster policies (Table 1 and Section A.4)
- Clarifying the concept of concurrent actions (Section 3.1 and new Figure 4).
- Describing concurrent knowledge representations in more depth (New Section A.3, new Figure 5)
- Adding an algorithm frame (New Algorithm 1)
- Contextualizing biologically-inspired related work with temporally-aware architectures (Section 2)

We believe our responses to Reviewer 3’s high-level concerns may potentially be useful to the other reviewers, so are including our responses here:

“1. Theoretically relatively straight forward”
We believe that the simplicity of our contribution is a feature, not a bug. We wanted to provide the simplest extension to existing image-based DRL implementations and measure the impact of making RL delay-aware. The simplicity of the implementation allows any discrete-time RL framework to be easily extended to handle delays by simply changing the network architecture inputs.

“2. Are not expressive enough to capture the problem in its full generality”
We believe that time-continuous solutions that generalize to more unconstrained problem settings are very promising future extensions, but outside the scope of this work. We add revisions to the manuscript to reflect this.

“3. Need more empirical justification with problems where their modification is indeed indispensable”
We believe that concurrent knowledge models are indispensable in speed critical vision-based real robot grasping tasks. A metric such as picks per minute are an important task for both research and practical use cases, and depends on policy speed as well as grasp success. As the reviewer suggested previously, optimizing for policy speed during reward-shaping is one baseline approach, but our experiments show that even with reward-shaping alone we reach an upper bound bottleneck on picks per minute that we were only able to surpass with concurrent action models.

---

### Decision · Program_Chairs · 2019-12-19

**Decision:**

Accept (Poster)

**Comment:**

This paper studies the setting in reinforcement learning where the next action must be sampled while the current action is still executing. This refers to continuous time problems that are discretised to make them delay-aware in terms of the time taken for action execution. The paper presents adaptions of the Bellman operator and Q-learning to deal with this scenario.

This is a problem that is of theoretical interest and also has practical value in many real-world problems. The reviewers found both the problem setting and the proposed solution to be valuable, particularly after the greatly improved technical clarity in the rebuttals. As a result, this paper should be accepted.